# SCF^FBW7-mediated degradation of Brg1 suppresses gastric cancer metastasis

Li-Yu Huang[1,2,3], Junjie Zhao[2,4], Hao Chen [4], Lixin Wan[2,5], Hiroyuki Inuzuka[2,6], Jianping Guo[2], Xuhong Fu[1], Yangyang Zhai[1], Zhaoning Lu[1], Xuefei Wang[4], Ze-Guang Han[1,3], Yihong Sun[4] & Wenyi Wei [2]

Brg1/SMARCA4 serves as the ATPase and the helicase catalytic subunit for the multi-component SWI/SNF chromatin remodeling complex, which plays a pivotal role in governing chromatin structure and gene transcription. However, the upstream signaling pathways regulating Brg1 protein stability and its physiological contribution to carcinogenesis remain largely elusive. Here we report that Brg1 is a bona fide ubiquitin substrate of SCF^FBW7. We reveal that CK1δ phosphorylates Brg1 at Ser31/Ser35 residues to facilitate the binding of Brg1 to FBW7, leading to ubiquitination-mediated degradation. In keeping with a tumor suppressive role of FBW7 in human gastric cancer, we find an inverse correlation between FBW7 and Brg1 expression in human gastric cancer clinical samples. Mechanistically, we find that stabilization of Brg1 in gastric cancer cells suppresses E-cadherin expression, subsequently promoting gastric cancer metastasis. Hence, this previously unknown FBW7/Brg1 signaling axis provides the molecular basis and the rationale to target Brg1 in FBW7-compromised human gastric cancers.

[1] Key Laboratory of Systems Biomedicine (Ministry of Education) and Collaborative Innovation Center of Systems Biomedicine, Shanghai Center for Systems Biomedicine, Shanghai Jiao Tong University, 800 Dong Chuan Road, 200240 Shanghai, China. [2] Department of Pathology, Beth Israel Deaconess Medical Center, Harvard Medical School, Boston, MA 02215, USA. [3] Shanghai-MOST Key Laboratory for Disease and Health Genomics and Key Laboratory of Systems Biomedicine (Ministry of Education), Chinese National Human Genome Center at Shanghai, 250 Bi Bo Road, 201203 Shanghai, China. [4] Department of General Surgery, Zhongshan Hospital, General Surgery Research Institute, Fudan University, 200032 Shanghai, China. [5] Department of Molecular Oncology, H. Lee Moffitt Cancer Center and Research Institute, Tampa, FL 33612, USA. [6] Center for Advanced Stem Cell and Regenerative Research, Tohoku University Graduate School of Dentistry, Sendai 980-8575, Japan. These authors contributed equally: Li-Yu Huang, Junjie Zhao. Correspondence and requests for materials should be addressed to L.-Y.H. (email: huangly@sjtu.edu.cn) or to Y.S. (email: sun.yihong@zs-hospital.sh.cn) or to W.W. (email: wwei2@bidmc.harvard.edu)

Brg1, also known as SMARCA4, encodes an ATPase subunit of the SWI/SNF chromatin remodeling complex, which can shift the position of nucleosomes by using the energy derived from ATP hydrolysis[1–4]. In keeping with an important role for the SWI/SNF chromatin remodeling complex in tumorigenesis, *Brg1* is frequently mutated or deleted in various types of human cancers including non-small-cell lung cancer and ovarian small cell carcinoma[5–8]. Notably, in these cancer types, mutations in *Brg1* display loss of function phenotypes and accordingly, Brg1 appears to function as a tumor suppressor in these tissue settings. However, the physiological role of Brg1 in tumorigenesis is rather complicated, and seems to be tissue type and cellular context dependent. For example, in pancreatic cancer setting, like the reported role of TGFβ signaling pathway[9,10], Brg1 exhibited both tumor-suppressive and oncogenic roles at distinct stages of pancreatic cancer formation, showing a cellular context-dependent manner[11,12].

On the other hand, Brg1 was significantly overexpressed in other human cancer types including breast cancer, medulla-blastoma and acute leukemia[13–16]. More importantly, in keeping with the oncogenic role for Brg1 in these cancer types, Brg1 was found to be essential for promoting cancer cell proliferation, and clinically high expression of Brg1 were correlated with poor outcome[13–16]. In these cancer types, Brg1 regulated a different set of gene expression from those in non-small-cell lung cancers[16]. In the gastric cancer setting, Sentani et al. observed no genetic mutations, but increased expression of Brg1 in 38 tumor samples[17]. Furthermore, relatively high Brg1 expression associated with the advanced stage and lymph node metastasis of gastric carcinoma[17]. These results indicate a possible oncogenic role for Brg1 in the gastric cancer setting. However, additional investigation is warranted to explore mechanistically how Brg1 protein is timely regulated and how aberrant elevation in Brg1 expression and oncogenic function facilitate gastric tumorigenesis.

Gastric cancer, as an aggressive form of disease in the gastric tract, remains the fourth most common cancer and the second leading cause of cancer-related death worldwide[18]. Peritoneal and distant metastasis have been considered invariably fatal situations of gastric cancer, and overall survival time of these patients were only 3–6 months[19] with no targeted therapies available. Thus, understanding the molecular mechanism that drives the metastasis event in gastric cancer becomes more imperative and significant, which may provide the molecular basis to design novel targeted therapy for this deadly disease.

To this end, the expression of *FBW7*, a bona fide tumor suppressor and a substrate recognition subunit of the SCF$^{FBW7}$ E3 ubiquitin ligase complex[20], was found to be decreased in gastric cancer at mRNA levels[21,22]. Moreover, low expression of FBW7 in primary gastric cancer contributed to tumor metastasis and poor prognosis[21,22]. More importantly, our mass-spectrometry-based screening indicated Brg1 as a putative substrate of FBW7[23]. In support of Brg1 functioning as a potential downstream effector that promote epithelial mesenchymal transition (EMT) and metastasis phenotypes in *FBW7*-compromised cells, re-expression of Brg1 was reported to repress E-cadherin and induce an EMT in pancreatic and colon cancers[12,24]. Hence, in this study, we further explored whether Brg1 overexpression in gastric cancer is in part due to *FBW7* reduction or loss and mechanistically how the FBW7/Brg1 signaling axis contributes to tumor metastasis and poor outcome of gastric cancer patients.

## Results
### Brg1 is an ubiquitin substrate of the SCF$^{FBW7}$ E3 ligase complex. By utilizing immunoprecipitation-based mass

spectrometry screenings[23], we have previously identified a number of FBW7-interacting proteins (like NFκB2, MYC and MAX) and some putative interactors of FBW7 in 293T cells. Among these FBW7-binding proteins, Brg1 (SMARCA4) was listed as one of the top candidates ($p = 0.021$)[23]. To further validate whether Brg1 is a downstream ubiquitin substrate of FBW7, we first examined Brg1 protein abundance changes in two *FBW7* knockout cell lines compared to the wild-type (WT) counterpart cells: *FBW7$^{-/-}$* DLD1 versus WT-DLD1 and *FBW7$^{-/-}$* HCT116 versus WT-HCT116 cells. Notably, we found that Brg1, but not its family members Arid1a and BRM, was elevated in *FBW7* depleted DLD1 and HCT116 cells (Fig. 1a and Supplementary Figure 1a), in which, c-Myc and Cyclin E, two well-characterized canonical FBW7 substrates, were used as positive controls[25,26]. We then examined the mRNA levels of Brg1 in these cell lines and observed no significant difference after depletion of *FBW7* in both cell lines (Supplementary Figure 1b). Moreover, the half-life of Brg1 was significantly extended in *FBW7$^{-/-}$* cells, and MG132 treatment resulted in increased Brg1 protein abundance (Fig. 1b–d), indicating a posttranslational regulation mode of Brg1 by FBW7.

We next investigated the relationship of Brg1 and FBW7 in human gastric cancer cell lines and found that Brg1 expression was inversely correlated with the expression of FBW7 (Supplementary Figure 1c). We further depleted *FBW7* in gastric cancer cell lines MKN45 and AGS, both of which express wild-type Brg1 and FBW7 according to the COSMIC (Catalogue of somatic mutations in cancer) cell line mutation analysis[27,28]. In keeping with Brg1 being as a possible ubiquitin substrate of FBW7, shRNA-mediated depletion of *FBW7* in MKN45 and AGS cells led to a marked elevation in protein abundance of endogenous Brg1, due to an increase in the half-life of endogenous Brg1 (Fig. 1e, f and Supplementary Figure 1d), whereas the mRNA levels of Brg1 were not altered (Supplementary Figure 1e). These data suggested that FBW7 could negatively regulate Brg1 protein stability in gastric cancer cells. In further support of this notion, we found that Brg1 could specifically bind to Cullin-1, but not other Cullin family members in cells (Fig. 1g). As a result, depletion of endogenous *Cullin-1* in MKN45 and AGS cells also led to an elevation of Brg1 protein abundance (Fig. 1h and Supplementary Figure 1f).

More importantly, phenocopying other known FBW7 ubiquitin substrates, Brg1 specifically interacted with wild type, but not the cancer-derived mutant forms of FBW7 (R465H, R479Q, R505C)[29,30] (Fig. 1i). Endogenous co-IP also confirmed the interaction between Brg1 and wild-type FBW7 in gastric cancer cells, in which, one of SWI/SNF subunit BAF155 were used as positive control (Fig. 1j and Supplementary Figure 1g). These mutants occurred in WD40 domain of FBW7, which have profound impact on substrate-binding affinity of FBW7[20]. In keeping with this result, re-introduction of wild type, but not the mutant forms of FBW7, led to dramatic decrease in protein abundance of FBW7 ubiquitin substrates including Brg1, c-Myc and Cyclin E (Fig. 1k).

There are three FBW7 isoforms identified in human genome (α, β, and γ)[20], to determine if there is any specificity for FBW7 isoforms to interact with, and to promote Brg1 ubiquitination, we examined the expression of three isoforms in gastric cancer cells and revealed FBW7α as the most abundant isoform (Supplementary Figure 2a). Importantly, Brg1 only bound to FBW7α, but not FBW7β or FBW7γ (Supplementary Figure 2b). Furthermore, ectopic expression of FBW7α, but not FBW7β or FBW7γ, suppressed Brg1 expression in *FBW7$^{-/-}$* cells (Supplementary Figure 2c). These results suggest that FBW7α is the dominant isoform of FBW7 that binds to and negatively regulates Brg1 protein stability.

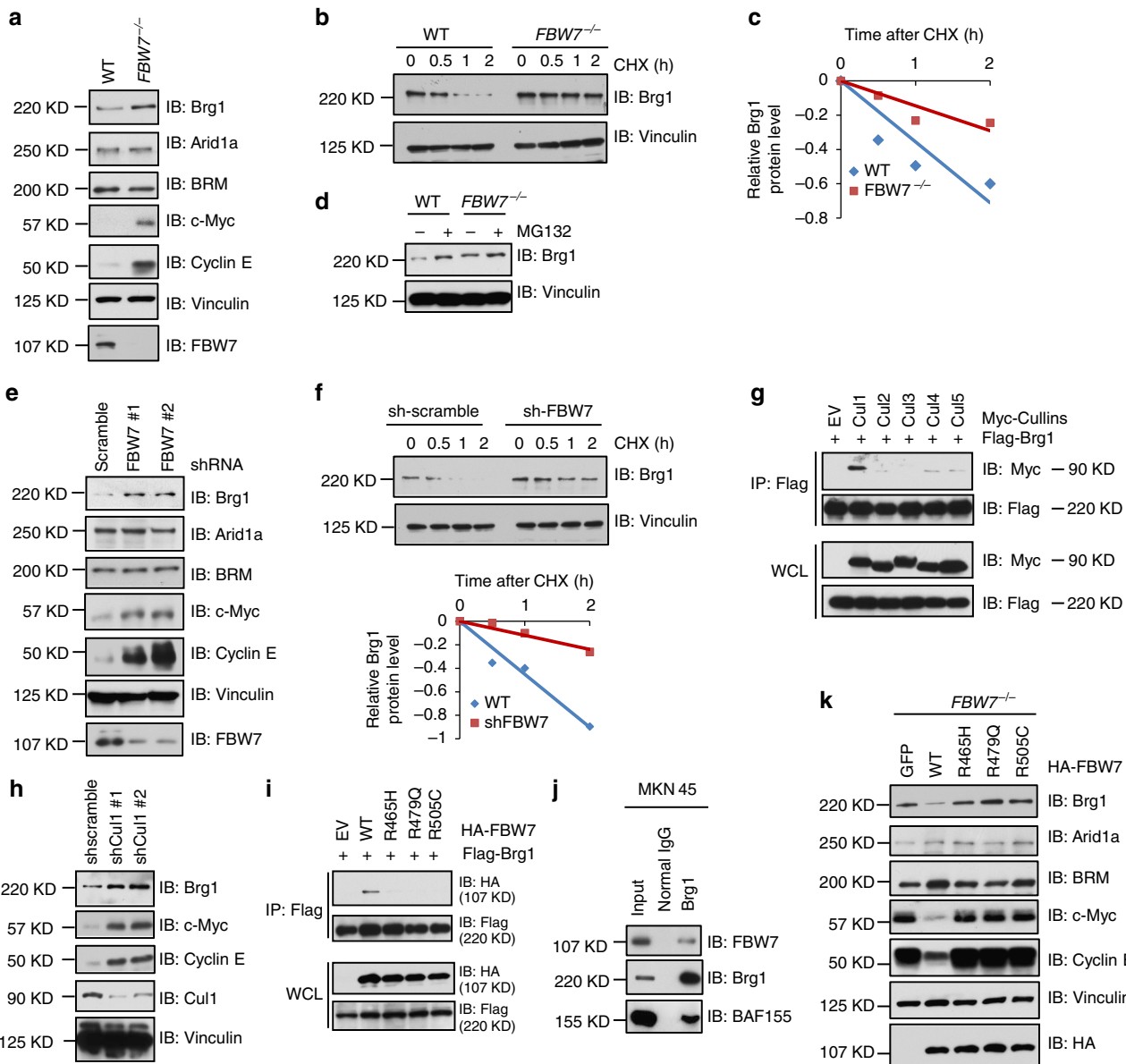

**Fig. 1** FBW7 negatively regulates the stability of Brg1. **a** Immunoblot analysis (IB) of whole cell lysates (WCLs) derived from wild-type (WT) and *FBW7*$^{-/-}$ DLD1 cells. **b**, **c** WT and *FBW7*$^{-/-}$ DLD1 cells were treated with 20 μg/ml cycloheximide (CHX). At the indicated time points, WCLs were prepared and IB analysis was carried out with indicated antibodies (**b**). The relative Brg1 intensity was normalized to Vinculin and then normalized to the $t = 0$ controls (**c**). **d** IB analysis of WCLs derived from WT and *FBW7*$^{-/-}$ DLD1 cells treated with MG132 (10 μM) or DMSO for 10 h. **e** IB analysis of WCLs derived from MKN45 cells infected with the indicated shRNA lentiviruses. **f** IB analysis of WCLs derived from MKN45 cells infected with the indicated shRNA lentiviruses and treated with 20 μg/ml CHX for indicated time periods. The relative intensity of Brg1 was normalized to Vinculin and then normalized to the $t = 0$ control (Bottom). **g** IB analysis of WCLs and immunoprecipitated (IPs) derived from 293T cells transfected with Flag-Brg1 together with the indicated constructs of Myc-tagged Cullin family members. **h** IB analysis of WCLs derived from MKN45 cells infected with the indicated lentiviral shRNA-*Cullin1* constructs. **i** IB analysis of WCLs and IPs derived from 293T cells transfected with Flag-Brg1 together with the indicated FBW7 constructs. **j** Co-IP experiments in MKN45 cells were performed using anti-Brg1 antibody (sc-17796, Santa Cruz). Mouse IgG was used as a control. **k** IB analysis of WCLs derived from *FBW7*$^{-/-}$ DLD1 cells infected with the indicated FBW7-expressing lentiviral vectors

**Phosphorylation of Brg1 by CK1δ is required for its ubiquitination by FBW7.** FBW7 substrates typically contain the FBW7 degradation motif (degron) that include phosphorylated serine or threonine residues in the "0" and "+4" positions[31–33]. Careful examination of the Brg1 protein sequence revealed the presence of only one evolutionarily conserved FBW7-recognizable degron in Brg1 sequence as shown in Fig. 2a. Hence, we constructed Brg1 mutants displacing Ser31 and Ser35 sites with alanine and found that mutation of the Ser31/Ser35 sites of Brg1 led to a sharp

reduction in interaction between Brg1 and FBW7 in cells (Fig. 2b), indicating that the Ser31/Ser35 sites phosphorylation is important for Brg1 to bind FBW7.

Next we sought to determine which upstream kinase is responsible for the phosphorylation of Brg1 and subsequent destruction of Brg1 by FBW7. We screened a panel of kinases commonly involved in FBW7 substrates phosphorylation and found that CK1δ, but not other kinases we examined, specifically promoted FBW7-mediated Brg1 destruction in 293T cells

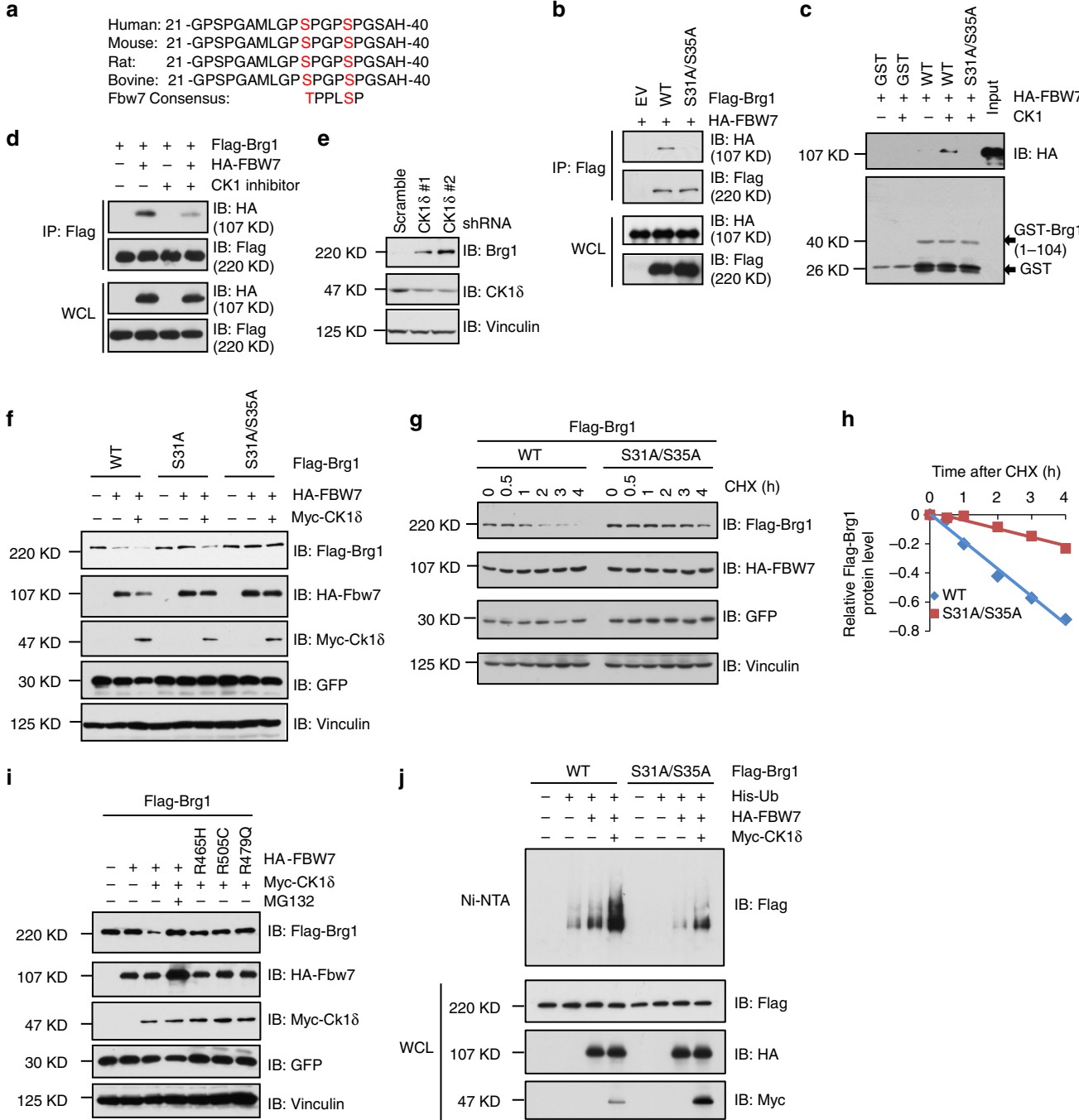

**Fig. 2** Brg1 interacts with FBW7 in a CKIδ-mediated phosphorylation dependent manner. **a** Sequence alignment of Brg1 with the phospho-degron sequences recognized by FBW7. The putative FBW7 phospho-degron sequence present in Brg1 is conserved across different species. **b** IB analysis of WCLs and IPs derived from 293T cells transfected with HA-FBW7 together with the indicated Flag-Brg1 constructs. **c** IB analysis showing the enhanced binding of HA-FBW7 with bacterially purified GST-Brg1 (1–104aa) proteins after incubation with CK1 kinase in vitro. GST was used as a negative control. **d** IB analysis of WCLs and IPs derived from 293T cells transfected with HA-FBW7 and Flag-Brg1. Where indicated, the CK1 inhibitor IC261 (20 μM) was added for 8 h before harvesting for IB analysis. DMSO was used as a negative control. **e** IB analysis of WCLs derived from MKN45 cells infected with the indicated lentiviral shRNA-*CK1δ* constructs. **f** IB analysis of WCLs derived from 293T cells transfected with the indicated Flag-Brg1 and HA-FBW7 vectors in the presence or absence of Myc-CK1δ. GFP was used as a negative control. **g, h** 293T cells were transfected with the indicated Flag-Brg1 constructs together with HA-FBW7 and Myc-CK1δ plasmids. Twenty hours after transfection, cells were split into 60-mm dishes. After another 20 h, cells were treated with 20 μg/ml CHX. **i** IB analysis of WCLs derived from 293T cells transfected with Flag-Brg1 and the indicated HA-FBW7 vectors. Where indicated, MG132 (10 μM) was treated for 10 h. **j** IB analysis of His tag pull-down and WCLs derived from 293 cells transfected with indicated Flag-Brg1 constructs together with the HA-FBW7, Myc-CK1δ, and His-Ub plasmids. 20 h after transfection, cells were treated with MG132 for 10 h before cell collection. His tag pull-down was then performed. Ni-NTA: nickel-nitrilotriacetic acid, Ub: ubiquitin

(Supplementary Figure 3a). Further studies revealed that other CKI family members are also capable, although with relatively weaker capacity, of promoting FBW7-mediated degradation of Brg1 in 293T cells (Supplementary Figure 3b). In keeping with this finding, we found that Brg1 specifically bound to CK1δ with higher affinity than with other CK1 family members in 293T cells (Supplementary Figure 3c). In vitro kinase reaction followed by pull-down analyses further revealed that CK1-mediated phosphorylation of Brg1 likely induced the interaction between FBW7 and WT, but not S31A/S35A mutant form of Brg1 in vitro (Fig. 2c). Consistently, CK1δ inhibitor treatment led to decreased interaction between Brg1 and FBW7 (Fig. 2d). Moreover, in keeping with a pivotal role for CK1δ in governing Fbw7-mediated ubiquitination of Brg1, depletion of endogenous CK1δ in gastric cancer cells MKN45 and AGS, resulted in a marked increase in protein abundance of endogenous Brg1 (Fig. 2e and Supplementary Figure 3d).

Furthermore, we found that the FBW7-non-interacting S31A/S35A mutant form of Brg1 was largely resistant to FBW7- and CK1δ-mediated destruction in 293T cells (Fig. 2f). Consistently, the half-life of S31A/S35A Brg1 was prolonged compared to that of wild-type Brg1 (Fig. 2g, h). Meantime, FBW7- and CK1δ-mediated Brg1 degradation could be blocked by the 26S proteasome inhibitor MG132 (Fig. 2i), while the substrate-binding mutants of FBW7 failed to promote the degradation of Brg1 in 293T cells (Fig. 2i). In further support of the essential role of the phospho-degron in FBW7-mediated degradation of Brg1, FBW7/CK1δ-triggered polyubiquitylation status of mutant Brg1 (S31A/S35A) was significantly reduced compared to that of WT Brg1 in 293T cells (Fig. 2j and Supplementary Figure 3e). These collective data coherently suggests that FBW7, as the substrate receptor for the SCF$^{FBW7}$ ubiquitin E3 ligase complex, binds to phosphorylated species of Brg1 and subsequently promotes Brg1 ubiquitination and degradation in a degron-dependent manner. During this process, our results further pinpointed CK1δ as the essential upstream kinase of Brg1 that phosphorylates Brg1 at Ser31/Ser35 residues to trigger its recognition by FBW7.

Next, we sought to determine the stressed or physiological conditions that could trigger Brg1 phosphorylation and subsequent ubiquitination mediated by FBW7/CK1δ (Supplementary Figure 4). We found that after etoposide or cisplatin treatment, the Brg1 expression was significantly decreased, while depleting FBW7 or CK1δ could rescue DNA damage-induced reduction in Brg1 protein abundance in MKN45 cells (Supplementary Figure 4a, d, e). Consistently, under etoposide or cisplatin treatment, the endogenous binding of FBW7 with Brg1 and Brg1 ubiquitination was significantly increased compared to normal conditions (Supplementary Figure 4f, g). On the other hand, Serum starvation or Nocodazole arrest and release seemed to have minimal effect on the expression of Brg1 (Supplementary Figure 4b, c).

**Brg1 expression inversely correlates with FBW7 to govern gastric tumorigenesis.** FBW7 is reported to display decreased mRNA expression in gastric cancer when compared to non-cancerous tissues[21]. Moreover, low FBW7 expression, but not FBW7 mutation status, was found to be associated with poor prognosis of clinical stage in gastric cancer patients[21]. Thus, we first examined FBW7 expression in 400 pairs of our human gastric cancer samples. Strikingly, we found that both FBW7 protein and mRNA levels decreased in examined 400 gastric cancers compared to the adjacent normal tissues (Fig. 3a, b). Consistent with a previous report[21], FBW7 expression was negatively associated with tumor progression and poor outcome

of gastric cancer patients (Supplementary Table 1 and Supplementary Figure 5a, b).

Next, we examined Brg1 mRNA and protein expressions in these clinical samples. Intriguingly, Brg1 protein, but not its mRNA levels were remarkably increased in cancer tissues compared to the adjacent normal tissues (Fig. 3b, c and Supplementary Table 2). Although Brg1 was once reported to have higher mRNA expression in advanced stage of 38 pairs of gastric cancer samples[17], in our case, we did not observe a significant difference between cancer tissues and normal tissues with respect to the mRNA level of Brg1. To gain further insight into the reasons of such difference apart from the different sample size (400 vs 38), we further analyzed Brg1 mRNA expression according to different stages and found that the T/N ratio of Brg1 expression in higher stage (TNM stage III–IV, T/N ratio = 1.59 ± 1.61, n = 237) is slightly elevated compared to that in the lower stage (TNM stage I–II, T/N ratio = 1.31 ± 1.47, n = 163) of gastric samples (p = 0.081). This result might indicate different mechanisms regulating Brg1 mRNA expression among different stages. But the total mRNA expression of Brg1 in 400 gastric cancers has no significant difference compared to normal tissues according to the statistical results (Fig. 3b).

Relatively high Brg1 protein abundance was positively correlated with tumor progression of these gastric cancer patients (Fig. 3d and Supplementary Figure 5c). Further analyses showed that high Brg1 protein level was positively associated with vascular invasion, lymph node metastasis and distant metastasis (Fig. 3e, Supplementary Figure 5d and Supplementary Table 2), as well as relatively shorter survival time of the gastric cancer patients (Fig. 3f).

In addition, IHC analyses revealed that the protein level of Brg1 was inversely correlated with FBW7 expression in gastric cancer specimens (Fig. 3g, h), indicating that Brg1, as an ubiquitin substrate of FBW7, was upregulated in gastric cancer in part due to decreased FBW7 expression that subsequently resulted in compromised enzymatic activity of the SCF$^{FBW7}$ E3 ligase complex. These data further suggested that Brg1, as a downstream oncogenic substrate of FBW7, might participate in the promotion of tumor progression and tumor metastasis caused by reduced FBW7 expression in gastric cancer setting. In support of this hypothesis, we observed an inverse correlation of Brg1 and E-cadherin expression, a tumor suppressive protein that has been reported to be suppressed by Brg1[12,24], in gastric cancer clinical samples (Fig. 3g, i).

In keeping with the finding that CK1δ is a crucial upstream kinase to phosphorylate Brg1 and to trigger Brg1 degradation mediated by FBW7, we further analyzed the correlation between Brg1 expression level and CK1δ activation in gastric cancer samples. The immunofluorescence result showed that the Brg1 level was inversely correlated with the activation of CK1δ in gastric tumor tissues (Supplementary Figure 6a, b).

**Brg1 promotes metastasis in gastric cancer cells.** We then analyzed the biological function of Brg1 in gastric cancer cell lines MKN45 and AGS. Notably, depleting endogenous Brg1 in MKN45 cells with relatively high levels of Brg1 to begin with (Supplementary Figure 1c), led to a elevation of E-cadherin level (Fig. 4a and Supplementary Figure 7). As a result, Brg1-depleted cells displayed attenuated cell migration, which can be reversed by further depletion of FBW7 (Fig. 4b–d, and Supplementary Figure 8a). However, there were no significant change in cell proliferation and colony formation capability when Brg1 was knocked down in MKN45 cells (Supplementary Figure 8b, c), suggesting that Brg1 mainly governs MKN5 cell migration rather than cell growth property.

To further assess the physiological role of Brg1 in promoting tumor metastasis, we employed tail-vein injection assay to observe long-distance tumor metastasis in vivo. To this end, we injected MKN45 cells into lateral tail vein of nude mice and checked metastatic nodules in lung tissues after 2 months. Notably, in comparison to control MKN45 cells, the metastatic nodules as well as the lung weight were significantly reduced in the mice injected with *Brg1*-depleted MKN45 cells, while additional depletion of *FBW7* in *Brg1*-depleted MKN45 cells could largely abrogate the observed reduction in metastasis in the lungs of nude mice (Fig. 4e–g).

In contrast, re-introduction of wild-type Brg1 into *Brg1*-depleted AGS cells led to a marked decrease of the endogenous E-cadherin level that was coupled with an elevation of Vimentin expression (Fig. 4h), indicative a possible epithelial to mesenchymal transition (EMT) phenotype[34,35]. In line with this

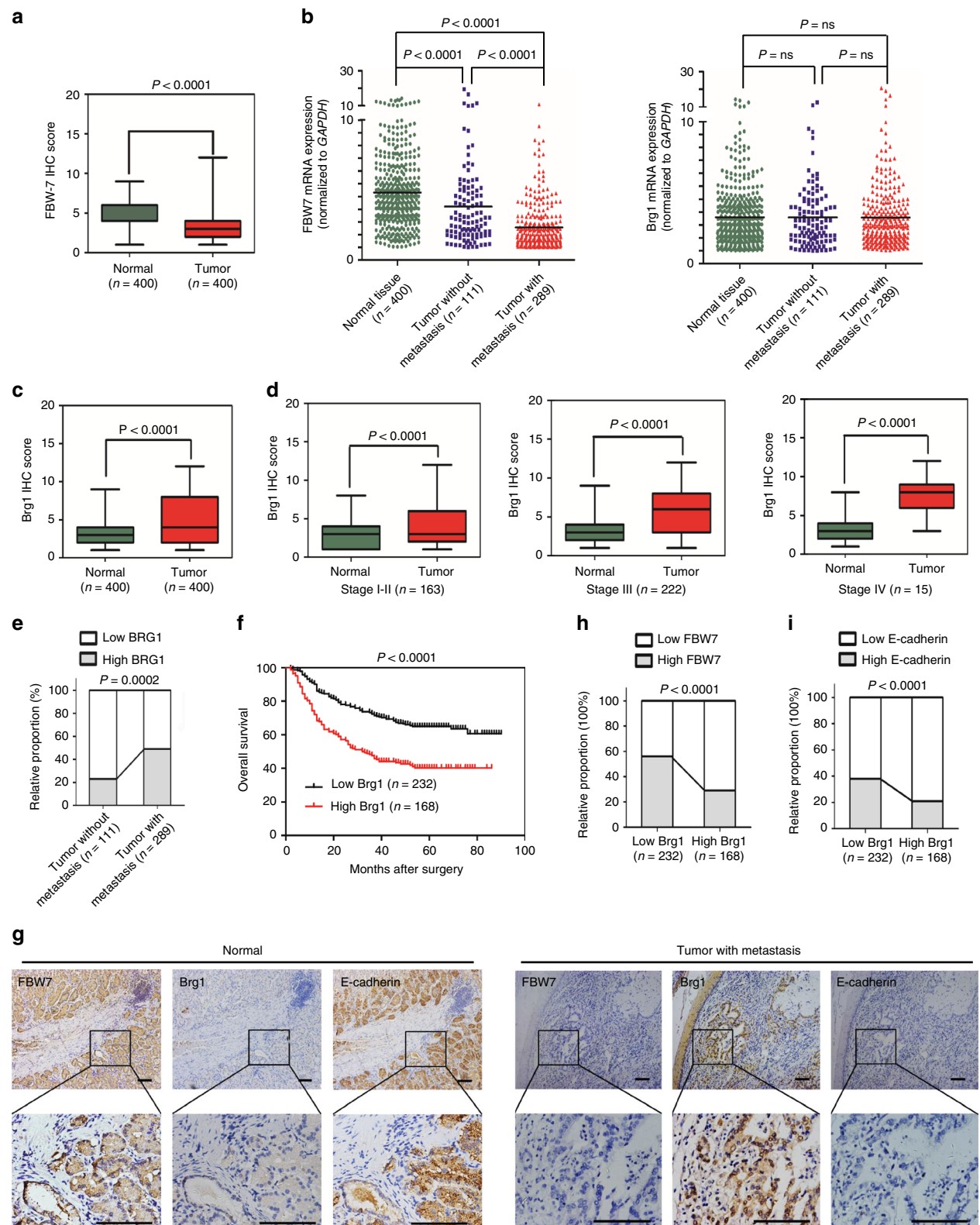

observation, cell migration and lung metastasis were significantly increased in WT-Brg1 reintroduced AGS cells (Fig. 4i–n, Supplementary Figure 8d). However, there were no significant alterations in cell growth curve and colony formation capability when Brg1 was ectopically expressed in AGS cells (Supplementary Figure 8e-f). Furthermore, compared to ectopic expression of wild type Brg1, expressing the FBW7-non-interacting mutant form of Brg1 (S31A/S35A) induced relatively more migration and metastasis phenotypes in AGS cells (Fig. 4i–n), presumably due to relatively higher expression levels via escaping FBW7-mediated degradation.

As well characterized previously, FBW7 could also promote the degradation of c-Jun and KLF-2, the two important transcription factors that regulate tumor cell proliferation and invasion[36,37]. Therefore, we also intended to figure out whether FBW7 negatively regulates tumor metastasis through targeting c-Jun or KLF-2 in gastric cancer settings. To this end, transwell results showed that although the depletion of *c-Jun* or *KLF-2* could downregulate the cell migration of MKN45 cells, but could not attenuate the increased cell migration induced by *FBW7* knockdown in MKN45 cells (Supplementary Figure 9a-f).

Furthermore, we examined the association of Brg1 expression and EMT phenotype in clinical patient samples, which was regularly defined by the collective staining of high Vimentin and low E-cadherin expression[38]. We found that Brg1 expression was positively correlated with the expression level of Vimentin (Supplementary Figure 10a, b), while inversely associated with E-cadherin expression (Fig. 3g, i and Supplementary Figure 10a).

These data collectively revealed that Brg1 could promote metastasis in gastric cancer cells, which can be antagonized by the FBW7 tumor suppressor largely through an ubiquitination-mediated degradation mechanism.

**Brg1 promotes EMT-driven gene transcription to induce metastasis**. To address the molecular mechanisms by which Brg1 contributes to gastric cancer cell migration and tumor metastasis, we examined a series of well-characterized EMT regulators such as Snail, ZEB-1, Twist-1 and β-catenin in AGS in a doxycycline (DOX)-based Brg1 inducible system (Fig. 5a). Interestingly, we found that Snail, but not other EMT-related transcription factors we examined including ZEB-1 and Twist, was notably increased upon induced Brg1 expression (Fig. 5a). Meanwhile, the epithelial marker E-cadherin was significantly reduced, while the mesenchymal marker Vimentin was remarkably increased (Fig. 5a), indicative of an EMT phenotype. Real-time PCR results also confirmed that EMT markers and transcription factor Snail were induced by ectopic expression of Brg1 (Fig. 5b). More interestingly, the change of cellular appearance towards a more elongated and spindle-shaped form was observed in AGS cells when Brg1 was induced after DOX treatment (Fig. 5c).

In contrast, when endogenous *Brg1* was depleted in MKN45 cells, expression of Snail and other EMT markers expression were significantly reduced (Fig. 5d, e). Moreover, when Brg1 was knocked down in *FBW7*[−/−] cells, the expression of Snail and EMT markers E-cadherin and Vimentin was restored to levels

comparable as *FBW7*[+/+] cells (Supplementary Figure 11a, b). On the other hand, reintroducing WT-Brg1, and to a much greater extent, the degron mutant Brg1 (S31A/S35A) in *Brg1*-depleted *FBW7*[+/+] cells, led to a sharp increase in Snail and Vimentin levels that were coupled with a marked reduction of E-cadherin (Supplementary Figure 11c).

Together, these results suggested that Brg1 might drive metastatic-supportive transcriptions in part through regulating Snail in gastric cancer cells and *Snail* depletion could significantly reduce the increased cell migration induced by Brg1 expression (Supplementary Figure 12a-c). To address whether Brg1 functions as a trans-activator by direct binding to the promoter of Snail, we performed a ChIP-PCR assay in MKN45 cells. Significantly, the snail promoter (−194/+59, respect to the transcription start)[39] could be immunoprecipitated by endogenous Brg1 when using anti-Brg1 antibody, whereas shRNA against endogenous *Brg1* abolished this binding (Fig. 5f, g). We also found the degron mutant form of Brg1 (S31A/S35A) could bind more to the snail promoter than the WT in CHIP assay by using Flag-antibody, which might be in part due to the higher expression level of Flag-Brg1 in degron-mutant form (Fig. 5h, i). Moreover, in *FBW7*[−/−] cells, which express relatively high level of Brg1 (Fig. 1a, b), an increased binding of Brg1 with Snail promoter was observed (Supplementary Figure 13a), while no binding of FBW7 with Snail promoter was detected (Supplementary Figure 13b). Moreover, compared to WT cells, a number of epigenetic markers such as H3K4me3, H3K27me3, H3K36me3 were increased in *FBW7*[−/−] cells, indicating the involvement of FBW7/Brg1 in chromosome remodeling[1] (Supplementary Figure 13c).

Our results therefore coherently suggest a model that Brg1 functions as an oncogene in the gastric cancer setting, and aberrancy in upstream signaling pathway, frequently being reduced Fbw7 expression in gastric cancer might predispose tumorigenesis and metastasis in part by elevating the Brg1/Snail oncogenic signaling axis. Our work further suggests Brg1 as a potential drug target in treating gastric cancers, especially those patients with compromised Fbw7 expression. In keeping with this notion, we found that Brg1 inhibitor PFI3 (SIGMA, SML0939), which targets the bromodomain in Brg1 and inhibits its binding with chromatin[40], could significantly suppress the lung metastasis of gastric cancer cell MKN45 in the tail-vein injection mice model (Fig. 5j–l).

## Discussion

In this work, we uncovered that Brg1, one of the two ATPase catalytic subunits of SWI/SNF family, was a bona fide ubiquitin substrate of the E3 ligase SCF[FBW7]. The SWI/SNF chromatin remodeling complexes were reported to use the energy of ATP hydrolysis to alter the nucleosome structure and regulate transcriptions. Here, we showed that Brg1 also acted as a trans-activator through binding to the promoter of EMT relevant genes such as the *Snail* gene and promote EMT and tumor metastasis in gastric cancer cells (Fig. 5). In doing so, Brg1 functions as a chromosome modifier, affecting several gene sets associated with

**Fig. 3** Brg1 expression inversely correlates with FBW7 to govern gastric cancer progression. **a** The scores of FBW7 immunohistochemical (IHC) staining in 400 tumor sections and paired adjacent non-tumor tissues. **b** The mRNA expressions of FBW7 and Brg1 were examined by real-time PCR. The data were analyzed by Student's *t* test. **c, d** The scores of Brg1 immunohistochemical staining in tumor sections and paired adjacent non-tumor tissues in total (**c**) or in different TNM (tumor node metastasis) stages (**d**). The data were analyzed by Student's *t* test. **e** High Brg1 expression positively correlated with lymph node metastasis in gastric cancer. Their correlation was analyzed by Spearman rank correlation test. **f** The association of Brg1 expression with overall survival was examined by the Kaplan–Meier analysis in gastric cancer patients from the Zhongshan cohort. **g** Representative images of FBW7, Brg1, and E-cadherin staining were shown in consecutive normal sections and distant metastatic tumor sections. Scale bar, 100 μm. **h, i** The correlations between Brg1 and FBW7 levels (**h**), Brg1 and E-cadherin levels (**i**) in 400 consecutive tumor sections that were semiquantified as high or low and analyzed by Spearman rank correlation test

cell migration and cell mobility[41]. During the peer-review of this paper, a report about FBW7 directly regulating the ubiquitination and proteolysis of Snail in NSCLC (non-small-cell lung cancer) cells[42] prompted us to further analyze which plays more critical role in the EMT regulation by FBW7. In Supplementary Figure 12b and c, we could observe the increased migration induced by Brg1 overexpression was significantly rescued by *Snail*

knockdown, but not to the normal level (EV group), which means except Snail, other Brg1 targets including the reported E-cadherin that participate in regulating cell migration and tumor metastasis in the gastric cancer setting requires further exploration.

By analyzing 400 pairs of human gastric cancer specimen, we found that Brg1 was highly expressed in tumor tissues, and inversely correlated with the tumor suppressor function of FBW7.

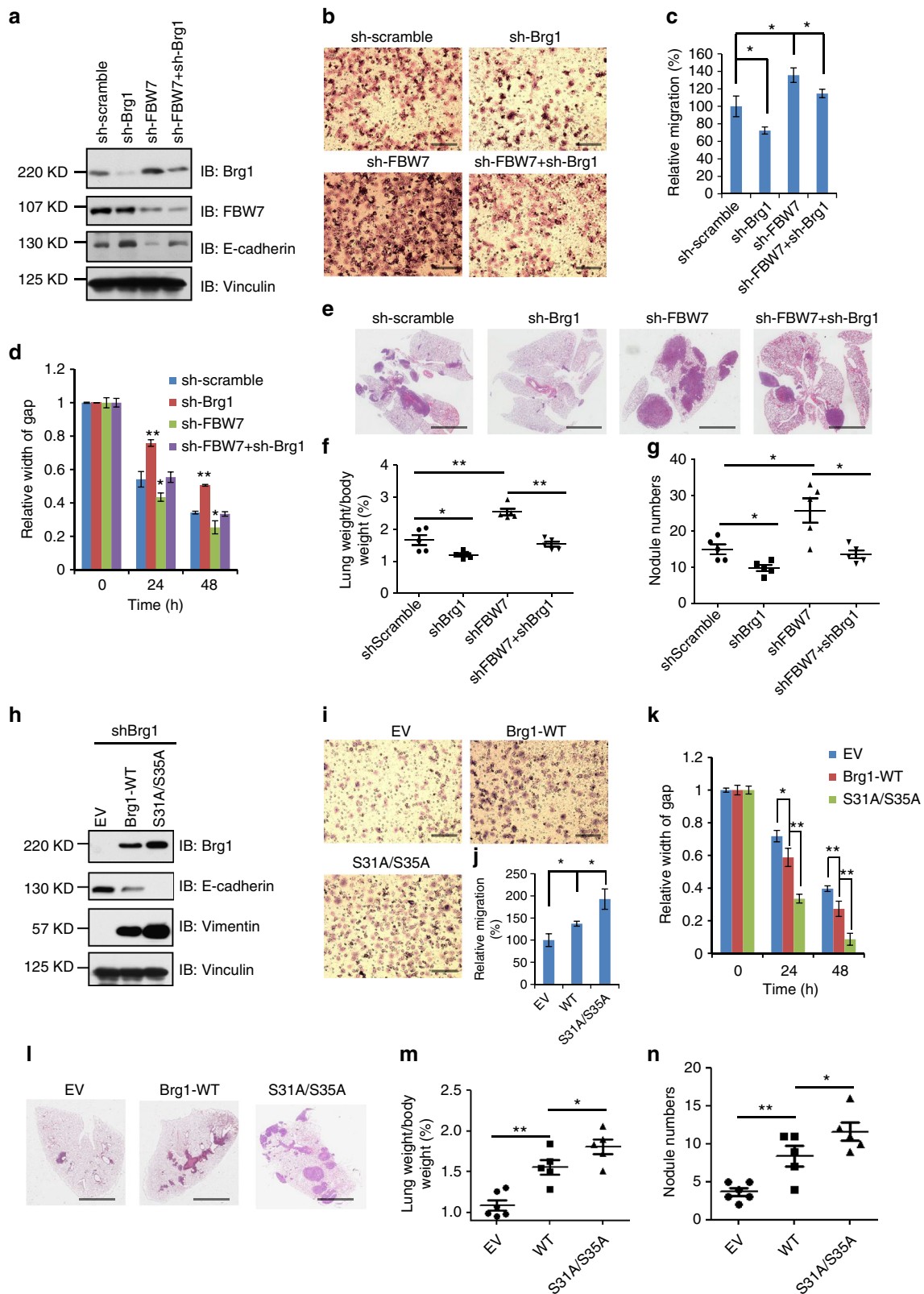

High Brg1 expression was associated with tumor progression, metastasis and poor outcome of gastric cancer patients, indicating a possible oncogenic role for Brg1, at least in the gastric cancer setting. However, it remains largely unknown how Brg1 protein stability is controlled in cells and whether aberrancy in Brg1 protein stability control contributes to tumorigenesis in gastric cancer setting. In this study, we also showed that FBW7 promoted CK1δ-mediated Brg1 ubiquitination and degradation. Furthermore, the phosphorylation of Brg1 Ser31/Ser35 sites, within the FBW7 degron consensus motif, enhanced the interaction between Brg1 and FBW7 and thus accelerated the ubiquitination of Brg1 by SCF$^{FBW7}$.

Therefore, Brg1 accumulation due to reduced FBW7 expression in gastric cancer could be one of the underlying molecular mechanism driving EMT supportive transcriptions which subsequently promotes tumor progression and metastasis, resulting in poor survival of gastric cancer patients (Fig. 6). Hence, our studies provide the molecular basis and the rationale for targeting the Brg1 oncoprotein as an effective therapeutic approach to treat gastric cancer patients with FBW7 deficiency.

## Methods

**Patient samples.** In total, 400 pairs of gastric cancer specimens were provided by the Department of General Surgery, Zhongshan Hospital (Fudan University, Shanghai, China). Normal mucosa tissues were taken from sites that were >50 mm away from primary lesions. Tissues were preserved in liquid nitrogen immediately for subsequent extraction of total RNA by Trizol reagent (Invitrogen) after the validation of pathological diagnosis using H&E staining of paraffin sections. Brg1 antibody (sc-17796, Santa Cruz) at a 1:50 dilution, FBW7 antibody (ab109617, Abcam) at a 1:1000 dilution, E-cadherin antibody (sc-8426, Santa Cruz) at a 1:50 dilution and Vimentin antibody (10366-1-AP, Proteintech) at a 1:300 dilution were used to interact with the dewaxed paraffin sections of the 400 pairs of gastric cancer samples. The use of human tissue samples and clinical data was approved by the ethics committee of Zhongshan Hospital. All donors were informed of the aim of the study and gave consent to donate their samples.

**Antibodies.** Anti-c-Myc antibody (SC-40, 1:1000), polyclonal anti-HA antibody (SC-805, 1:1000), anti-Cullin-1 antibody (SC-70895, 1:1000), and anti-Cyclin E antibody (SC-247, 1:1000) were purchased from Santa Cruz. Polyclonal anti-FLAG antibody (F2425, 1:1000), monoclonal anti-FLAG antibody (F-3165, 1:1000), anti-Vinculin antibody (V9131, 1:5000), peroxidase-conjugated anti-mouse secondary antibody (A4416, 1:3000), peroxidase-conjugated anti-rabbit secondary antibody (A4914, 1:3000) and CK1 inhibitor IC261 were purchased from Sigma. Anti-Myc-tag (2272, 1:1000), anti-GST (2625, 1:1000), anti-Vimentin (5741, 1:1000), anti-Snail (3879, 1:1000), anti-ZEB-1 (3396, 1:1000), anti-Twist1 (4119, 1:1000), anti-β-catenin (9582, 1:1000), anti-Arid1a (12354, 1:1000) and anti-BRM (11966, 1:1000) antibodies were purchased from Cell Signaling. Anti-FBW7 antibody (A301-720A, 1:1000) was purchased from Bethyl. Monoclonal anti-HA antibody (MMS-101P, 1:1000) was purchased from Covance. Anti-GFP antibody (632380, 1:1000) was purchased from Invitrogen.

**Cell culture.** MKN45 and AGS cells were from the Gastric Cancer Center of Fudan University. Human embryonic kidney 293 (HEK293) cells, HEK293FT, Wild type and *FBW7*$^{-/-}$ DLD1, and Wild type and *FBW7*$^{-/-}$ HCT116 were maintained in Wenyi Wei lab. All the cell lines were cultured in Dulbecco's Modified Eagle's Medium (DMEM) containing 10% fetal bovine serum (FBS), 100 units of penicillin

and 100 mg/ml streptomycin. All cell lines were routinely tested to be negative for mycoplasma contamination.

**Plasmids.** HA-FBW7 was obtained from Dr. Keiich I Nakayama (Kyushu University, Japan). HA-ERK1 was obtained from Dr. John Blenis (Weill Cornell Medical College, NY). Myc-CK1 was obtained from Dr. J Wade Harper (Harvard Medical School, MA). HA-GSK3β and Myc-Cullin family constructs were obtained from Dr. James A DeCaprio (Dana-Farber Cancer Institute, MA). Flag-IKKα, Flag-IKKβ, and HA-NIK were purchased from Addgene. FBW7 and Brg1 mutants were generated using a QuikChange XL Site-Directed Mutagenesis Kit (Stratagene, La Jolla, CA), according to the manufacturer's instructions.

**Immunoprecipitation assay.** Cells were lysed in EBC (50 mM Tris pH 8.0, 120 mM NaCl, 0.5% NP-40) buffer supplemented with protease inhibitors (Complete Mini, Roche) and phosphatase inhibitors (phosphatase inhibitor cocktail set I and II, Calbiochem). For immunoprecipitation, 1 mg lysates were incubated with the appropriate antibody (1–2 μg) for 3–4 h at 4 °C followed by 1 h incubation with Protein-A Sepharose beads (GE Healthcare). Immuno-complexes were washed four times with NETN buffer (20 mM Tris, pH 8.0, 100 mM NaCl, 1 mM EDTA and 0.5% NP-40) before resolved by SDS-PAGE for immunoblotting. Quantification of the immunoblot band intensity was performed with Image J software.

**In vitro ubiquitination assay.** 293T cells were transfected with His-Ub and the desired constructs. Twenty-four hours post transfection, cells were treated with 10 mM MG132 for 10 h. Cells were lysed in buffer A (6 M guanidine-HCl, 0.1 M Na$_2$HPO$_4$/NaH$_2$PO$_4$ and 10 mM imidazole pH 8.0) and sonicated. The lysates were incubated with nickel-nitrilotriacetic acid (Ni-NTA) matrices (QIAGEN) for 3 h at room temperature. The His pull-down products were washed twice with buffer A, twice with buffer A/TI (1 volume buffer A and 3 volumes buffer TI), and one time with buffer TI (25 mM Tris-HCl and 20 mM imidazole pH 6.8). The pull-down proteins were resolved by SDS-PAGE for immunoblotting.

**shRNA synthesis.** The pLKO lentiviral vectors to knockdown CUL1 (#1: sense, 5′-GATTTGATGGATGAGAGTGTA-3′; #2: sense, 5′-GCCAGCATGATCTCCAA GTTA-3′) were obtained from the Dana Farber/Harvard Cancer Center DNA Resource Core. The shRNA sequences of pLKO vectors to deplete FBW7 are: #1: sense, 5′-AACCTTCTCTGGAGAGAGAAA-3′; #2: sense, 5′- CCAGAGAC TGAAACCTGTCTA-3′. The pLKO shCK1 and shGFP constructs were obtained from Dr. J Wade Harper (Harvard Medical School, MA). The pLKO-lentiviral vectors to deplete endogenous BRG1 (#1: TRCN0000015551, #2: TRCN0000015552) were from OpenBiosystems. The corresponding lentiviruses were packaged and generated by transfecting each pLKO lentiviral plasmid with pCMV-VSV-G and pCMV-dR8.9 into 293T cells using a polyethyenimine (PEI) transfection protocol.

**Protein half-life detection.** Cells were plated in six-well plates at 80% confluence and cultured in Dulbecco's modified Eagle's medium (Invitrogen, Carlsbad, CA) with 10% fetal bovine serum. After 24 h of culture, cycloheximide (CHX) (20 μg/ml, Sigma) was added. Cells were lysed in EBC buffer containing protease and phosphatase inhibitor (Sigma) after CHX treatment at the indicated time points.

**In vitro invasion assay.** Cell migration was measured using Transwell inserts (8 μm pores, Corning Life Sciences). Cells (1.5 × 10$^4$) were suspended in 100 μl serum-free medium and added on the membrane. The 600 μl medium supplemented with serum was used as a chemo attractant. Cells that migrated after 12 h were fixed in 0.4% PFA, stained with 0.5% crystal violet, and counted using an inverted microscope. For each independent experiment, three replicates per condition were run.

**Fig. 4** Brg1 promotes cell migration in vitro and metastasis in vivo in gastric cancer cells. **a** IB analysis of WCLs derived from MKN45 cells infected with the indicated lentiviral shRNA constructs. **b**, **c** Representative images of migrated MKN45 cells infected with the indicated lentiviral shRNA constructs in a transwell assay (**b**) and quantification of migrated cells in (**c**). Data were shown as mean ± SD of three independent experiments. *$p < 0.05$, Student's $t$ test. Scale bar, 100 μm. **d** The migration of MKN45 cells transfected with the indicated lentiviral shRNA plasmids in scratch experiments. Data were shown as mean ± SD of three independent experiments. *$p < 0.05$, **$p < 0.01$, Student's $t$ test. **e–g** Representative images and statistical analysis of metastasis formed in lungs of each nude mouse ($n = 5$) at 2 months after tail vein injection of MKN45 cells (3 × 10$^6$) with the indicated lentiviral shRNA plasmids. Scale bar, 1 mm. **h** IB analysis of whole cell lysates derived from AGS cells transfected with the indicated Flag-Brg1 constructs. **i**, **j** Representative images of migrated AGS cells transfected with the indicated Flag-Brg1 constructs in a transwell assay and quantification of migrated cells. Data were shown as mean ± SD of three independent experiments. *$p < 0.05$, Student's $t$ test. Scale bar, 100 μm. **k** The migration of AGS cells transfected with the indicated Flag-Brg1 constructs in scratch experiments. Data are shown as mean ± SD of three independent experiments. *$p < 0.05$, **$p < 0.01$, Student's $t$ test. **l–n** Representative images and statistical analysis of metastases that formed in lungs of each nude mouse ($n = 5$) at 2 months after tail vein injection of AGS cells (2 × 10$^6$) transfected with the indicated Flag-Brg1 plasmids. *$p < 0.05$, **$p < 0.01$, Student's $t$ test. Scale bar, 1 mm

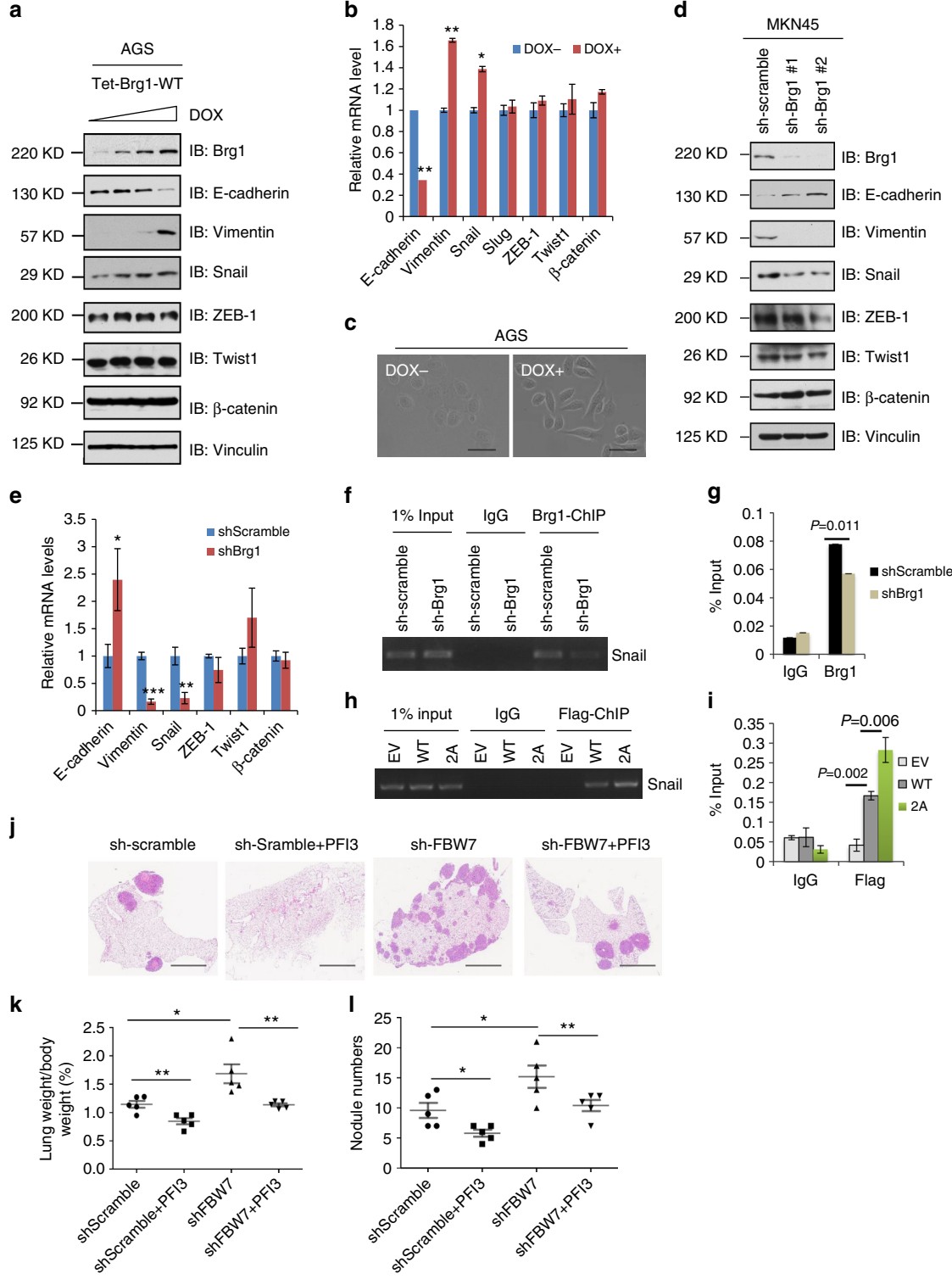

**In vitro scratch assay**. Cells were plated in 60 mm dish. The cell monolayer was scraped in a straight line with a P200 pipet tip. Photographs of the scratch were taken at 0, 24, and 48 h. Gap width at 0 h was set to 1. Gap width analysis was performed with Image J software. Measurements were taken at multiple defined sites along the scratch. Each scratch was given an average of all measurements. Data are expressed as the average of three independent experiments.

**In vivo metastasis assay**. MKN45 and AGS cells transduced with the indicated lentiviruses or plasmids were resuspended in growth media. Each mouse was

injected with $3 \times 10^6$ cells (MKN45) or $2 \times 10^6$ cells (AGS) and was randomly allocated to each group. To determine the appropriate group sizes for the different experiments, the sample size calculation tool from the University of Iowa was used (http://homepage.stat.uiowa.edu/~rlenth/Power/index.html). Six-week-old Nude mice were purchased from Shanghai Laboratory Animal Center, Chinese Academy of Sciences. Mice were fed with a regular diet (RD) and had free access to water and food. All the mice experiments were approved by the Animal Ethics Committee of the Shanghai Model Organisms Center. After 2 months, mice were sacrificed and collected the lung tissues in order to observe the metastasis by biotechnician in double blind. Pictures of the same lung were stitched and macroscopic metastases

**Fig. 5** Brg1 promotes metastasis in gastric cancer by driving an EMT transcriptional program. **a** IB analysis of WCLs derived from gastric cancer cell lines AGS. AGS cells were infected with the indicated pTRIPZ-Brg1 lentiviral vectors and were selected with 1 μg/ml puromycin for 72 h to eliminate non-infected cells. Afterward, 100, 300, 1000, 3000 ng/ml doxycycline (DOX) was added to the cells for another 24 h before harvesting for IB analysis. **b** Real-time PCR assays were performed in pTRIPZ-Brg1 AGS cells in the present or absence of DOX (1000 ng/ml) for 24 h. *$p < 0.05$, **$p < 0.01$, Student's $t$ test. **c** Representative images of the cell morphology in AGS cells. Scale bar, 100 μm. **d** IB analysis of WCLs derived from MKN45 cells infected with the indicated lentiviral shRNA plasmids. **e** Real-time PCR assays were performed in MKN45 cells infected with the indicated lentiviral shRNA plasmids. *$p < 0.05$, **$p < 0.01$, Student's $t$ test. **f**, **g** ChIP assays with anti-Brg1 antibody were performed in MKN45 cells infected with the indicated lentiviral shRNA plasmids. Mouse IgG was used as negative control. **h**, **i** ChIP assays with anti-Flag antibody were performed in AGS cells transfected with the indicated Flag-Brg1 constructs. Mouse IgG was used as antibody control in this experiment. **j–l** Representative images and statistical analysis of metastases that formed in lungs of each nude mouse ($n = 5$) at 2 months after tail vein injection of MKN45 cells ($3 \times 10^6$/each) transfected with the indicated lentiviral shRNA plasmids. One milligrams PFI3 (SIGMA, SML0939) dissolved in dimethyl sulfoxide plus 0.25 ml phosphate-buffered saline was given intraperitoneally 3 times per week. *$p < 0.05$, **$p < 0.01$, Student's $t$ test. Scale bar, 1 mm

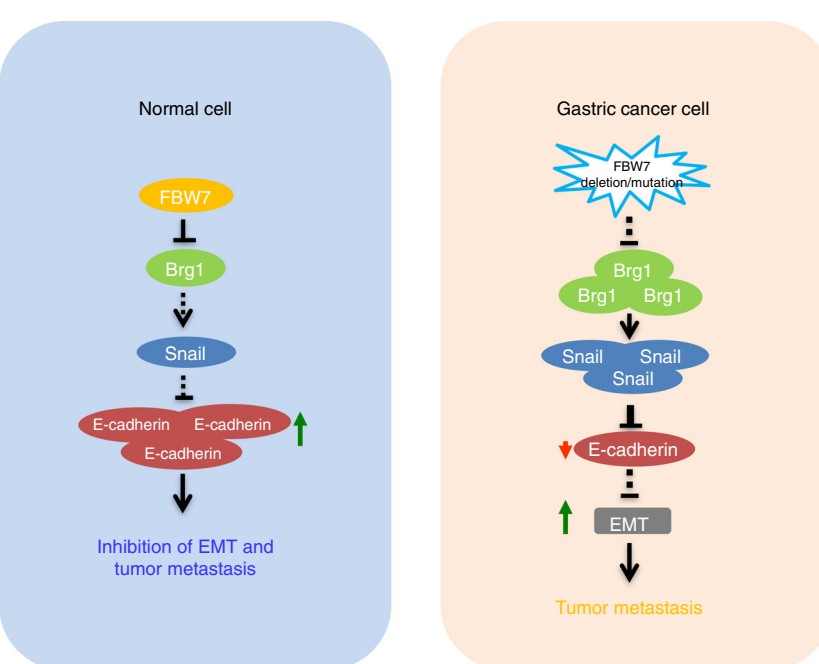

**Fig. 6** The proposed model for FBW7 in suppressing metastasis in gastric cancer setting. The FBW7/Brg1 signaling axis governs gastric cancer EMT changes and plays a critical role in gastric cancer metastasis

were quantified by counting lesions in both sides of the lung per mouse. Representative images from tumors from 5 mice per condition are included. Data were plotted using GraphPad PRISM and significance was determined by unpaired *t*-test.

**Cell viability assays**. Cells were plated in 96-well plates ($1 \times 10^3$) and the cell growth was monitored by absorbance using the MTS assay according to the manufacturer's instructions (Promega) at the indicated time. Cell growth was measured in a microplate reader. Experiments were repeated three times.

**Colony formation assays**. Cells were plated in six-well culture dishes (BD) at a density of 300 cells /well. Two weeks later, Cells were stained with crystal violet on the plates and counted. Experiments were repeated three times.

**Chromatin immunoprecipitation (ChIP)**. ChIP assay was performed using the EZ ChIP Kit (Millipore) according to the manufacturer's protocol.

**Statistical analysis**. Statistical evaluation of in vitro and in vivo experiments was calculated using Student's *t*-test. Multiple group comparisons were analyzed by one-way ANOVA. Kaplan–Meier survival curves and log-rank (Mantel–Cox) tests were used for survival analysis of patients with gastric cancer. All statistical analyses

were two-sided, different cutoff values, $P < 0.05$ (*), $P < 0.01$ (**), and $P < 0.001$ (***), were considered significant.

### Data availability
Gel source images for Figs. 1, 2, 4, 5 and Supplementary Figures 1–4, 9, 11–13 are available in Supplementary Figure 14. All the other data supporting the findings of this study are available from the corresponding authors upon reasonable request.

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

## Acknowledgements

pCMV5 BRG1-Flag was a gift from Joan Massague (Addgene plasmid # 19143), GFP-SMARCA4 was a gift from Kyle Miller (Addgene plasmid # 65391). W.W. and H.I. are American Cancer Society Research Scholar. This work was supported in part by the NIH grants to W.W. (R01GM089763 and CA229307) and the National Natural Science Foundation of China to L.-Y.H. (81402317, 81872346). L.-Y.H. is supported by a post-doctoral fellowship from China Scholarship Council.

## Author contributions

L.-Y.H. and W.W. designed the experiments; L.-Y.H. and J.Z. performed most of the experiments; Y.S., J.Z., H.C., X.W. collected and analyzed patient samples; L.W. designed most of the primers; L.W., H.I. and J.G. assisted in the analysis of some experiments. X.F., Y.Z. and Z.L assisted in some biochemical experiments; Z.-G.H. helped interpreting the data; W.W. and L.-Y.H. wrote the manuscript.

## Additional information

**Competing interests:** The authors declare no competing interests.

