## [Peer Review File · Nature Communications]

Reviewers' comments:

Reviewer #1 (Expertise: Ubiquitination, cancer, Remarks to the Author):

In this manuscript, Huang et al. reported that Brg1, a ATPase catalytic subunit of SWI/SNF family, is a novel substrate of SCFFbw7 E3 ubiquitin ligase via a series of biochemical assays. The authors further defined the biological significance of the FBW7-Brg1 axis in regulation of invasion and metastasis of gastric cancer, by acting as a tumor suppressor and oncogene, respectively, with support from a correlation study of 400 pairs of gastric cancer vs. adjacent normal tissues, showing reduced FBW7 and increased Brg1 levels in cancer tissues. Mechanistically, Brg1 trans-activates Snail to promote EMT and tumor metastasis.

Overall, this is a hypothesis-driven and straight forward study reporting a novel and significant finding with potential translational value. The experiments were largely well-designed and executed with the conclusion mainly supported by the results. However, the manuscript indeed suffers from several deficiencies, which are outlined below for authors to address, before it warrants a publication at NC.

Major concerns:

- 1) Most experiments were conducted under overexpression or knockdown conditions. The authors need to define under what physiological or stressed conditions that would trigger Brg1 phosphorylation and subsequent ubiquitylation by Fbw7 to really show the physiological or pathological relevance and significance of the Fbw7-Brg1 axis.
- 2) Since FBW7 also degrades other migration-associated onco-proteins, such as c-Jun (via slug) and KLF-2, how can the authors exclude the possibility that these Fbw7 substrates also play a role in EMT/migration of gastric cancer cells?

Minor concerns:

- 1) Figures 1&2. All the bindings tested between Brg1 and Fbw7 were under overexpressed conditions. Can two proteins bind to each other under endogenous/physiological conditions or triggered by pathological stresses?
- 2) Fig 2g, two panels (WT & S31A/S35A) should be run in the same gel to have fair comparison.
- 3) Fig 3. Given the fact that there is no good commercially available Fbw7 Ab, the authors should validate that the Fbw7 Ab they used for IHC study indeed detected Fbw7, but not other non-specific proteins. The use of Fbw7+/+. vs. Fbw7-/- HCT116 and DLD cells will help to resolve this issue.
- 4) Fig 4a, it seems unlikely that siFBW7 would completely eliminate E-cadherin expression, since it is an indirect effect via the Brg1-Snail axis. Thus, it is necessary to show potential change at the mRNA level of E-cadherin.
- 5) Fig 4i-m, it seems that mutated forms of Brg1 inhibited E-cadherin and promoted migration much more effectively than the WT. The authors' explanation was that the mutant escaped the degradation by FBW7. It will be more convincing to show by a ChIP assay that the mutant form indeed binds to the snail promoter, and is more effectively than the WT due to a higher level.
- 6) Fig 5f-g, For the ChIP assay, it will be much more convincing, if the authors would show the image of PCR band (not only the bar chart) to demonstrate the difference.

Reviewer #2 (Expertise: Gastric cancer, metastasis, Remarks to the Author):

This "SCFFBW7-mediated Degradation of Brg1 Suppresses Gastric Cancer Metastasis" is demonstrated that Brg1 is significantly up-regulated in gastric cancer and it was ubiquitinated by FBW7. The regulation of Brg1 by FBW7 is firstly reported in this manuscript, but they need to have novelty the special function of FBW7 in gastric cancer. Because, FBW7 (FBXW7) is well known tumor suppressor genes and low expression of FBW7 in primary gastric cancer contributed to tumor. FBW7 is also known to regulate EMT. They also need to more detail explain about the analysis of gastric patient data.

In this manuscript they determined FBW7 regulates EMT by Brg1 protein stability. Upon Zhang et al (Cancer Letter 2018), FBW7 directly regulates the ubiquitination and proteolysis of the transcriptional factor Snail and EMT phenotype. They need to prove the molecular mechanism, what is more critical function of FBW7 on EMT by regulation of Brg1 or snail.

Moreover, Upon Sentani K et al, expression level of Brg1 mRNA is upregulated with no genetic mutations. Furthermore, relatively high Brg1 expression associated with the advanced stage and

lymph node metastasis of gastric carcinoma. In this manuscript, they determined no change of mRNA expression between normal and malignant tissues, but, they didn't explain how they get the patient mRNA samples. They need to explain exactly and why they had the different data. They also determined the phosphorylation of Brg1 by CK1 δ is important to recognize by FBW7. It suggests that the protein stability is largely regulated by CK1 δ . Therefore, they should analysis the expression and phosphorylation of Brg1 from the level and activation of CK1 δ in malignant tissues of gastric patients.

Brg1 has also other main function for chromatin remodeling. Is FBW7 regulate chromatin remodeling by FBW7 in gastric cancer? It is also interesting study involving in this gastric cancer study.

Major point:

1. They need to determine the reverse expression between endogenous protein level of FBW7 and Brg1 in several gastric cancer cell lines.
2. Fbxw7 contains 3 isoforms (Fbxw7 α , Fbxw7 β , and Fbxw7 γ), and they are differently regulated in subtract recognition. Therefore, the authors need to clarify what type of FBW7 is dominant in gastric cancer and regulates Brg1.
3. The experiment of tail vein injection of cancer cells is not good for analysis of EMT. Because the cancer cells already in blood stream. Therefore, they need to find other experiment or staining in patient samples to determine EMT phenotype.

Reviewer #1

In this manuscript, Huang et al. reported that Brg1, a ATPase catalytic subunit of SWI/SNF family, is a novel substrate of SCFFbw7 E3 ubiquitin ligase via a series of biochemical assays. The authors further defined the biological significance of the FBW7-Brg1 axis in regulation of invasion and metastasis of gastric cancer, by acting as a tumor suppressor and oncogene, respectively, with support from a correlation study of 400 pairs of gastric cancer vs. adjacent normal tissues, showing reduced FBW7 and increased Brg1 levels in cancer tissues. Mechanistically, Brg1 trans-activates Snail to promote EMT and tumor metastasis.

Overall, this is a hypothesis-driven and straight forward study reporting a novel and significant finding with potential translational value. The experiments were largely well-designed and executed with the conclusion mainly supported by the results. However, the manuscript indeed suffers from several deficiencies, which are outlined below for authors to address, before it warrants a publication at NC.

Response: We really appreciate the professional comment and a positive evaluation on our study. In the revised version, in the light of the constructive comments of the referee, we further improve our manuscript.

Major concerns:

1) Most experiments were conducted under overexpression or knockdown conditions. The authors need to define under what physiological or stressed conditions that would trigger Brg1 phosphorylation and subsequent ubiquitylation by Fbw7 to really show the physiological or pathological relevance and significance of the Fbw7-Brg1 axis.

Response: We thank the reviewer for the constructive suggestions. According to the reviewer, we first examined the expression level of Brg1 protein under different stressed conditions (etoposide/cisplatin treatment; serum starvation; nocodazole synchronization). We found that only under DNA damage conditions (etoposide/cisplatin treatment), Brg1 levels were decreased sharply, while rescued by the depletion of *FBW7* or *CK1δ* (Supplementary Fig.4a-e). In keeping with this result, we further showed that the binding of Brg1 to FBW7 and the subsequent Brg1 ubiquitination were enhanced under DNA damage conditions (Supplementary Fig.4f-g). These results suggested that DNA damage triggered CK1δ-mediated Brg1 phosphorylation and subsequent ubiquitination by FBW7. These results were consistent with our previous findings showing ATM could phosphorylate CK1δ and activate CK1δ. (*Oncotarget* 2012; 3: 1026-1035). We described the newly obtained data on Page 10-11 of the revised manuscript.

2) Since FBW7 also degrades other migration-associated onco-proteins, such as c-Jun (via slug) and KLF-2, how can the authors exclude the possibility that these Fbw7 substrates also play a role in EMT/migration of gastric cancer cells?

Response: We thank the reviewer for the professional suggestions. In the revised manuscript, we added transwell assay in sh-c-Jun or sh-KLF-2 MKN45 cells. We found that the depletion of *c-Jun* or *KLF-2* down-regulated the cell migration of MKN45 cells, but could not abolish FBW7 loss-induced elevation of cell migration in MKN45 (Supplementary Fig. 9a-f), meantime *Brg1* depletion can largely reduce the increased cell migration of shFBW7 cells (Fig 4b,c). These newly obtained results indicated that at least in gastric cancer cells, Brg1 played a major role in the EMT/migration regulation by FBW7. We described the data on Page 15-16 of the revised manuscript.

Minor concerns:

1) Figures 1&2. All the bindings tested between Brg1 and Fbw7 were under overexpressed conditions. Can two proteins bind to each other under endogenous/physiological conditions or triggered by pathological stresses?

Response: We fully agree with the reviewer's critical points. In Fig. 1i, Supplementary Fig. 1g and Supplementary Fig. 4f of the revised version, we conducted endogenous co-IP by using Brg1 or FBW7 antibodies in MKN45 cells, which confirmed the interaction between endogenous Brg1 and endogenous FBW7 in gastric cancer cells. We also found the binding of Brg1 to FBW7 was enhanced under DNA damage conditions in Supplementary Fig. 4f. We described the data on Page 7 of the revised manuscript.

2) Fig 2g, two panels (WT & S31A/S35A) should be run in the same gel to have fair comparison.

Response: As kindly suggested by the reviewer, we repeated the experiment and ran both samples in the same gel as shown in Fig. 2g of the revised version.

3) Fig 3. Given the fact that there is no good commercially available Fbw7 Ab, the authors should validate that the Fbw7 Ab they used for IHC study indeed detected Fbw7, but not other non-specific proteins. The use of Fbw7+/+. vs. Fbw7-/- HCT116 and DLD cells will help to resolve this issue.

Response: As kindly suggested by the reviewer, we performed Immunofluorescence (IF) in WT versus *FBW7*-/- DLD1 cells by using FBW7 antibody (ab109617, Abcam) as shown in Supplementary Fig. 5a. The IF results validated the specificity of the FBW7 antibody (ab109617, Abcam).

4) Fig 4a, it seems unlikely that siFBW7 would completely eliminate E-cadherin expression, since it is an indirect effect via the Brg1-Snail axis. Thus, it is necessary to show potential change at the mRNA level of E-cadherin.

Response: As kindly suggested by the reviewer, we showed a longer exposure of E-cadherin band in Fig 4a and performed RT-PCR at the same time to monitor E-cadherin mRNA expression level. The data was shown in Supplementary Fig. 7.

5) Fig 4i-m, it seems that mutated forms of Brg1 inhibited E-cadherin and promoted migration much more effectively than the WT. The authors' explanation was that the mutant escaped the degradation by FBW7. It will be more convincing to show by a ChIP assay that the mutant form indeed binds to the snail promoter, and is more effectively than the WT due to a higher level.

Response: We appreciated the professional suggestions by the reviewer. We added ChIP assay in AGS cells transfected with WT Brg1 or mutant Brg1 (S31A/S35A) (Fig. 5h,i). The ChIP-PCR results showed that degen mutant form of Brg1 could bind more to the Snail promoter than WT Brg1 in part due to higher Brg1 expression via escaping FBW7-mediated degradation. We described the data on Page 18 of the revised manuscript.

6) Fig 5f-g, For the ChIP assay, it will be much more convincing, if the authors would show the image of PCR band (not only the bar chart) to demonstrate the difference.

Response: We appreciated the professional suggestions by the reviewer. We added the images of PCR band as shown in Fig. 5f, h and Supplementary Fig. 13.

Reviewer #2

This “SCFFBW7-mediated Degradation of Brg1 Suppresses Gastric Cancer Metastasis” is demonstrated that Brg1 is significantly up-regulated in gastric cancer and it was ubiquitinated by FBW7. The regulation of Brg1 by FBW7 is firstly reported in this manuscript, but they need to have novelty the special function of FBW7 in gastric cancer. Because, FBW7 (FBXW7) is well known tumor suppressor genes and low expression of FBW7 in primary gastric cancer contributed to tumor. FBW7 is also known to regulate EMT. They also need to more detail explain about the analysis of gastric patient data. In this manuscript they determined FBW7 regulates EMT by Brg1 protein stability. Upon Zhang et al (Cancer Letter 2018), FBW7 directly regulates the ubiquitination and proteolysis of the transcriptional factor Snail and EMT phenotype. They need to prove the molecular mechanism, what is more critical function of FBW7 on EMT by regulation of Brg1 or snail.

Response: We appreciate the professional comments of the reviewer as well as the reminding of new reference about the FBW7-Snail study in non-small cell lung cancer (NSCLS) during the reviewing of our paper.

In our study, we revealed that Brg1, as a downstream ubiquitin substrate of FBW7, might drive EMT in part through regulating Snail in gastric cancer cells. Brg1 could bind to the Snail promoter and drive metastatic-supportive transcriptions (Fig. 5). In addition to Snail, Brg1 was also reported to bind to E-cadherin promoter and promoted EMT in pancreatic cancer and colon cancer (*Oncogene 2010; Genes & development 2015*). In our gastric cancer studies, the inverse correlation of Brg1 with E-cadherin was also revealed in 400 gastric cancer samples (Fig. 3g, i and Supplementary Fig. 10). As well-known from previous studies, Brg1 functions as a chromosome modifier, affected several gene sets associated with cell migration and cell mobility (*Proc Natl Acad Sci U S A. 2013*). Therefore, we fully agree with the reviewer that more Brg1 targets that participate in regulating cell migration and tumor metastasis requires our further exploration, which we have discussed in the “Discussion” section (Page 20).

In addition, to experimentally address this raised concern, we included newly generated transwell assays in gastric cancer cell line AGS (Supplementary Fig. 12) by depleting Snail using shRNA constructs. The transwell results showed that the increased migration induced by Brg1 overexpression could be significantly rescued by further *Snail* depletion, but not recovered to the normal level (EV group), which means that in addition to Snail, Brg1 governs EMT partially through other mechanisms as analyzed above. We also cited the new reference from Zhang et al. (*Cancer Letter 2018*) and analyzed in the Discussion section (Page 20) in the revised manuscript.

Moreover, Upon Sentani K et al, expression level of Brg1 mRNA is upregulated with no genetic mutations. Furthermore, relatively high Brg1 expression associated with the advanced stage and lymph node metastasis of gastric carcinoma. In this manuscript, they determined no change of mRNA expression between normal and malignant tissues, but, they didn't explain how they get the patient mRNA samples. They need to explain exactly and why they had the different data.

Response: We appreciate the professional suggestions of the reviewer as well as for pointing out the difference of the analysis of Brg1 mRNA expression between normal and malignant tissue from our study and Sentani K et al report.

As kindly suggested by the reviewer, we added the detailed description of how we obtained the patient samples in Materials and Methods on Page 26. Furthermore, the clinical characteristics of these samples were also organized in Supplementary Table1/2.

Regarding the difference of the analysis of Brg1 mRNA expression from our study and Sentani K et al report, the huge different sample size (400 vs 38) might be one of the reasons. Beyond that, in Sentani K et al case, we noticed that only in advanced stage (stage III-IV), there was much higher Brg1 mRNA expression, while in lower stage tumors (stage I-II), they failed to see the difference. Therefore, we tried to dig out whether there were differences of Brg1 mRNA expression according to different stages. Notably, we found that the T/N ratio (mean \pm SD) of Brg1 expression in more advanced stages (TNM stage III-IV, T/N ratio=1.59 \pm 1.61, n=237) is slightly elevated compared to that in less advanced stages (TNM stage I-II, T/N ratio=1.31 \pm 1.47, n=167) of gastric samples (p=0.081), which might indicate different mechanisms regulating Brg1 mRNA

expression among different tumor stages. However, the total mRNA expression of Brg1 in 400 gastric cancers has no significant difference compared to normal tissues according to the statistical results (Fig. 3b). We discussed the difference on Page 12 in revised manuscript and fully agree with the reviewer that additional in-depth studies are warranted to further expand our understanding in this critical question. However, we hope the reviewer will agree that this lies outside the major scope of this manuscript and requires a following up study in a separate manuscript.

They also determined the phosphorylation of Brg1 by CK1 δ is important to recognize by FBW7. It suggests that the protein stability is largely regulated by CK1 δ . Therefore, they should analysis the expression and phosphorylation of Brg1 from the level and activation of CK1 δ in malignant tissues of gastric patients.

Response: As kindly suggested by the reviewer, we analyzed the correlation between Brg1 expression level and CK1 δ activation in gastric cancer tissues (Supplementary Fig. 6) through Immunofluorescence assay by using Brg1 antibody (sc-17796, Santa Cruz) and CK1 δ antibody (14388-1-AP, Proteintech). The immunofluorescence results showed that Brg1 level was inversely correlated with the activation of CK1 δ in gastric tumor tissues (Supplementary Fig. 6). However, since the specific Brg1 phosphorylation antibody has not been produced yet, we cannot see the correlation of Brg1 phosphorylation with CK1 δ activation in human samples of this current study. We described the data on Page 13 in the revised manuscript.

Brg1 has also other main function for chromatin remodeling. Is FBW7 regulate chromatin remodeling by FBW7 in gastric cancer? It is also interesting study involving in this gastric cancer study.

Response: We thank the reviewer for the professional suggestions. Indeed, Brg1 as the ATPase and helicase catalytic subunit of SWI/SNF complex, has main functions for chromosome remodeling. So, in *FBW7*^{-/-} cells, we also observed the changes of some histone modification like the elevation of H3K4Me3, H3K27Me3 and H3K36Me3 (Supplementary Fig. 13c). In the meantime, we also performed ChIP assay by using FBW7 antibody to see whether FBW7 could locate to the chromatin regions like Brg1. The ChIP-PCR result (Supplementary Fig. 13b) showed that FBW7 fails to bind to the Snail promoter. We described the data on Page 18 in revised manuscript.

Major point:

1. They need to determine the reverse expression between endogenous protein level of FBW7 and Brg1 in several gastric cancer cell lines.

Response: As kindly suggested by the reviewer, we examined the FBW7 and Brg1 expression in several gastric cancer cell lines as shown in (Supplementary Fig. 1c). The FBW7 status was defined according to the COSMIC (Catalogue of somatic mutations in cancer) cell line mutation analysis. The IB results showed that Brg1 expression was inversely correlated with the expression of FBW7 and Brg1 expression was relatively higher in FBW7-mutated cell lines MKN1 (*FBW7* R465C) and IM95 (*FBW7* R505C) (Supplementary Fig. 1c). We described the data on Page 6-7.

2. Fbxw7 contains 3 isoforms (Fbxw7 α , Fbxw7 β , and Fbxw7 γ), and they are differently regulated in subtract recognition. Therefore, the authors need to clarify what type of FBW7 is dominant in gastric cancer and regulates Brg1.

Response: As kindly suggested by the reviewer, we first performed RT-PCR analysis of FBW7 isoforms (α , β , γ) gene expression in gastric cancer cell lines MKN45 and AGS. The RT-PCR results showed that FBW7 α was the major type of FBW7 in gastric cancer cells (Supplementary Fig. 2a). Furthermore, we performed co-IP to see which isoform binds Brg1 better. The co-IP results showed that in MKN45 cells, only FBW7 α , but not FBW7 β or FBW7 γ binds to Brg1 specifically (Supplementary Fig. 2b). In keeping with this finding, we reintroduced the different FBW7 isoforms (α , β , γ) into *FBW7*^{-/-} cells and found that FBW7 α degrades Brg1 more efficiently (Supplementary Fig. 2c). We described the data on Page 8 of the revised manuscript.

3. The experiment of tail vein injection of cancer cells is not good for analysis of EMT. Because the cancer cells already in blood stream. Therefore, they need to find other experiment or staining in patient samples to determine EMT phenotype.

Response: We agree with the reviewer that tail vein injection model is not a direct method for analysis of EMT, but still a widely used and solid evidence to moderate the ability of metastasis (*Gastroenterology* 2016; *J. Natl. Cancer Inst* 2016; *Gut* 2017; *Oncogene* 2018; *Theranostics* 2018. etc.). As kindly suggested by the reviewer, we examined the association of Brg1 expression and EMT phenotype in clinical patient samples, which was regularly defined by the collective staining of high Vimentin and low E-cadherin expression (*Gut* 2013; *Hepatology* 2015; *J. Clin. Invest* 2014). Notably, we found that Brg1 expression was positively correlated with the expression level of Vimentin (Supplementary Fig. 10), while inversely associated with E-cadherin expression (Fig. 3g, i and Supplementary Fig. 10). We described the data on Page 16 in the revised manuscript.

REVIEWERS' COMMENTS:

Reviewer #1 (Remarks to the Author):

The authors have adequately addressed all critiques I raised previously, and I do not have any issues.

Reviewer #2 (Remarks to the Author):

They addressed and re-experimented carefully for responses. It was well revised.